# Influence of Natural Killer Cells and Natural Killer T Cells on Periodontal Disease: A Systematic Review of the Current Literature

**DOI:** 10.3390/ijms21249766

**Published:** 2020-12-21

**Authors:** Andreas Seidel, Corinna L. Seidel, Matthias Weider, Rüdiger Junker, Lina Gölz, Helga Schmetzer

**Affiliations:** 1Dental Practice, Bahnhofstraße 10, 82223 Eichenau, Germany; 2Department of Orthodontics and Orofacial Orthopedics, Universitätsklinikum Erlangen and Friedrich-Alexander Universität (FAU) Erlangen-Nürnberg, Glückstr. 11, 91054 Erlangen, Germany; matthias.weider@uk-erlangen.de (M.W.); lina.goelz@uk-erlangen.de (L.G.); 3Center for Dental Prosthetics and Biomaterials, Danube Private University Krems, Steiner Landstraße 124, 3500 Krems-Stein, Austria; ruediger.junker@dp-uni.ac.at; 4Department of Medical III, University Hospital LMU Munich, Marchioninistraße 15, 81377 Munich, Germany; helga.schmetzer@med.uni-muenchen.de

**Keywords:** natural killer cells, natural killer T cells, periodontitis, host defense, innate immunity, adaptive immunity

## Abstract

Natural killer (NK) cells, as members of the innate immune system, and natural killer T (NKT) cells, bridging innate and adaptive immunity, play a prominent role in chronic inflammatory diseases and cancerogenesis, yet have scarcely been examined in oral diseases. Therefore, systematic research on the latest literature focusing on NK/NKT cell-mediated mechanisms in periodontal disease, including the time period 1988–2020, was carried out in MEDLINE (PubMed) using a predetermined search strategy, with a final selection of 25 studies. The results showed that NK cells tend to have rather proinflammatory influences via cytokine production, cytotoxic effects, dendritic-cell-crosstalk, and autoimmune reactions, while contrarily, NKT cell-mediated mechanisms were proinflammatory and immunoregulatory, ranging from protective effects via B-cell-regulation, specific antibody production, and the suppression of autoimmunity to destructive effects via cytokine production, dendritic-cell-crosstalk, and T-/B-cell interactions. Since NK cells seem to have a proinflammatory role in periodontitis, further research should focus on the proinflammatory and immunoregulatory properties of NKT cells in order to create, in addition to antibacterial strategies in dental inflammatory disease, novel anti-inflammatory therapeutic approaches modulating host immunity towards dental health.

## 1. Introduction

Periodontitis is one of the most common chronic inflammatory diseases in humans and has been suggested as a risk factor for various systematic pathologies, such as diabetes mellitus [1] and coronary heart disease [2]. The bacterial etiology of the disease and the role of the host response in the pathogenesis have been examined in several investigations. Infection with gram-negative microorganisms is well-recognized as one primary etiologic factor for periodontal disease [3]. Interestingly, the prevailing understanding that specific ‘periopathogens’ of the ‘red complex’, namely *Porphyromonas gingivalis*, *Tanarella forsythia*, and *Treponema denticola*, are primarily responsible for the disease etiology has changed [4]. New models of the pathogenesis of periodontitis designate *Porphyromas gingivalis* as the keystone pathogen [5] orchestrating, rather than affecting, tissue damage and bone resorption, which is mediated by so-called ‘pathobionts’ [6]. Prominent members of these ‘pathobionts’ are *Aggregatibacter actinomycetemcomitans*, *Tanarella forsythia*, and *Treponema denticola*, which are supposed to be directed by the keystone pathogen *Porphyromas gingivalis*, initiating inflammatory reactions [7]. Finally, periodontal destruction is caused by excessive innate and adaptive immune responses developed against pathogenic bacteria, whereas bacterial plaque is considered to be one key etiologic factor [8]. The inflammatory immune response is estimated to account for almost 80% of the risk of periodontal tissue damage [9], leading to hard- and soft-tissue breakdown and clinical signs of periodontitis [10]. Recent studies contemplating the oral microbiome elucidated that periodontitis is a dysbiotic inflammatory disease. Alterations in the abundance or influence of individual microbial species causing dysbiosis of the oral microbiota affect host–microbe homeostasis, resulting in a destructive reaction involving the innate and adaptive branch of immunity [11,12].

Natural killer cells were found in 1975 by Rolf Kiessling [13]. They received their particular name due to their ability to kill tumor cells without prior stimulation or exposure to antigens [13]. Natural killer (NK) cells, as subsets of large granular lymphocytes, comprise approximately 10–15% of all lymphocytes and present one of the major parts of the innate, cell-mediated immune response. One distinct difference to natural killer T (NKT) cells is that they lack a rearranged T-cell receptor [14], but still feature receptors such as CD56, CD161, NKp46, CD94, CD16, and killer-cell immunoglobulin-like receptors [15,16]. NK cells are not only responsible for initiating an adaptive immune response, but also for regulating autoimmune mechanisms. The direct killing of pathogens by natural killer cells is affected via the production of perforin and granzyme [14]. Furthermore, NK cells can secret large amounts of interferon gamma similar to CD4^+^ T-helper cells (Type I) and cytotoxic T cells [17]. Their proinflammatory and/or immunoregulatory role in pathologies such as cancer, diabetes mellitus, and atherosclerosis has been depicted in previous studies [18]. Different types of cancer, including breast cancer, prostate cancer, decidual malformations [19,20,21], leukemia, and lymphoma [22,23], can be altered by NK cell functions by modifying tumor microenvironment components such as the tumor necrosis factor alpha, acidics, tumor exosomes, hypoxia, and adenosine [24]. This process leads to reduced tumor cell lysis by a decreased expression of activatory receptors and the release of perforin and granzyme [18]. Moreover, the production of interferon gamma of NK cells promotes the auto-aggressiveness of T cells in patients with Type I diabetes [25,26] and modifies the immune defense towards an excessive, uncontrolled, and unresolved response [27]. Conversely, Rodacki et al. [28,29] did not show any significant difference between the activatory state of NK cells taken from peripheral blood in individuals with Type I diabetes at different stages. Additionally, NK cells were found in an atherosclerotic plaque in humans and mice [30,31]. Several cytokines and chemokines, such as monocyte attractant protein (MCP-1) or fractaline (CX3CL1), can enhance NK cell migration towards atherosclerotic plaques, leading to the activation of NK cells and increased interferon gamma release [32,33]. Cytokines such as interleukin-12, -15, and -18, which are produced in atherosclerotic plaques, seem to attract natural killer cells and promote their crosstalk with other immune cells, including dendritic cells, macrophages, and monocytes [34,35,36]. Within atherosclerotic lesions, the crosstalk between NK cells and dendritic cells might worsen the progression of disease via interferon gamma release and the secretion of matrix metalloproteinases (MMPs), resulting in damage of the extracellular matrix and atherosclerotic plaque destabilization [37].

Natural killer T cells bridge innate and adaptive immunity [38]. They are a subset of T-lymphocytes showing characteristics specific for the T and NK cell family [39,40,41]. Like T cells, they possess a T-cell receptor (TCR) reacting—unlike T-lymphocytes—with lipid or glycolipid antigens presented by the major histocompatibility complex (MHC) class I-related glycoprotein CD1d [39,40,41]. Most NKT cells express a semi-invariant TCR consisting of Vα14-Jα18 and Vβ-8.2, -7, or -2 chains in mice or Vα24-Jα18 and Vβ11 chains in humans and are referred to as type I or invariant NKT (iNKT) cells [41]. There is also another NKT cell sublineage consisting of type II or variant NKT (vNKT) cells, which is characterized by expressing more diverse TCRs and playing a cross-regulating role with iNKT cells [42]. The biological functions of NKT cells unfold when their TCR is stimulated, leading to rapid and strong cytokine secretion and the acquisition of cytotoxic activity [43]. The major antigen directly activating NKT cells is α-linked galactosylceramide [44]. It is not possible to predict the impact of NKT cells on an immune response or disease, as they can exhibit pro- and anti-inflammatory properties in different diseases [45]. For instance, NKT cells display tolerogenic properties in graft-versus-host disease after bone marrow transplantation or hepatic allografts [46,47]. Furthermore, in patients with leukemia, antileukemic properties of NKT cells were observed by triggering dendritic cell-derived immune surveillance [48]. Type I or type II NKT cells can affect other cells, such as CD8^+^ T cells and NK cells, by an increased or decreased production of interferon gamma, contributing to more or less resistance towards tumor development [49]. Interestingly, NKT cells can worsen atherosclerosis and contact hypersensitivity by a boost of pathogenic cytokines and the activation of other cells [50]. Moreover, they can activate host antibodies, leading to pathological autoimmune reactions in arthritis and biliary cirrhosis [51,52].

Taken together, NK and NKT cells exert pro-inflammatory and immunomodulatory effects in different pathologies and might also play a significant role in host defense against bacterial invasions during dental inflammatory diseases such as periodontitis. However, modes of influence triggered by NK cells and NKT cells are difficult to predict in advance. Therefore, the aim of this article is to investigate the physiological functions of NK cells and NKT cells to reestablish oral health and/or the pathophysiological mechanisms of NK cells and NKT cells in periodontal disease.

## 2. Results

The PRISMA Statement was utilized as a reporting item [53] and a literature search of MEDLINE (PubMed) was carried out using a defined strategy (Section 4: Material and Methods). A search of the latest literature comprising the time period 1988–2020 resulted in 89 publications. After the application of distinct inclusion criteria, 56 of the records were excluded. Then, 33 of the records were screened for further relevance. Six papers out of 33 were not available. For the remaining 27 items, full-text articles were assessed. Two further full-text articles were excluded due to lacking relevance to NK cells and NKT cells and/or periodontitis. Finally, 25 of the studies were identified for qualitative analysis (Figure 1).

As mentioned above, 25 studies were included for examining the influence of NK cells and NKT cells on periodontal disease. The following tables (Table 1, Table 2 and Table 3) give an overview of the eligible literature. NK cells were the subject of 19 studies, NKT cells of four studies, and combined NK cells/NKT cells of two studies. An overview of the literature showed different study designs concerning the evaluation of NK cell- and NKT cell-mediated mechanisms in periodontitis. The authors performed human studies with biopsies and/or blood analysis, animal, and/or in vitro experiments with subsequent cell cultivation, immunostaining, or transcriptomic analysis. For the evaluation of a possible influence of NK and NKT cells on periodontal disease (Table 1, Table 2 and Table 3), the terms ‘proinflammatory’ and/or ‘immunoregulatory’ with the following definitions were utilized: 

Proinflammatory influence: In the cited study, NK cells and/or NKT cells directly or indirectly cause tissue destruction through different mechanisms, leading to the aggravation of disease;

Immunoregulatory influence: In the cited study, NK cells and/or NKT cells directly or indirectly prevent tissue destruction through the activation of mechanisms leading to the resolution of disease progression or through the suppression of mechanisms exacerbating damage;

Proinflammatory and immunoregulatory influence: In the cited study, NK cells and/or NKT cells show both aforementioned mechanisms;

Uncertain biological relevance: In the cited study, the biological relevance of NK cells and/or NKT cells in periodontitis needs further investigation. 

Moreover, a color code indicating mechanisms induced by NK and NKT cells in periodontitis was created (Figure 2). 

### 2.1. NK Cell-Mediated Mechanisms in Periodontitis

NK cell-mediated mechanisms were evaluated in nineteen different investigations, which are represented in the following overview (Table 1).

**Table 1 ijms-21-09766-t001:** NK cell-mediated mechanisms in periodontitis. NK cell, natural killer cell; DC–dendritic cell; A.a.—*Aggregatibacter actinomycetemcomitans*; P.g.—*Porphyromonas gingivalis*; F.n.—*Fusobacterium nucleatum*; LPS—lipopolysaccharide; TLR4—Toll-like receptor; IgG2—immunoglobulin G2; IL-2—interleukin 2; IFN-ɣ—interferon gamma; TNF—tumor necrosis factor; CRACC—CD2-like receptor-activating cytotoxic cell; NCR-1, natural cytotoxicity triggering receptor 1; LFA-1—Leukocyte-function-associated antigen-1; KIR—killer-cell immunoglobulin-like receptor; CD—cluster of differentiation; gF—gingival fibroblast; and PB—peripheral blood.

Mechanism	Evaluation of the Influence	Study Design	Reference
NK cells may participate in local responses through cytotoxic and/or immunoregulatory mechanisms.	Proinflammatory through tissue damage	Human biopsies Cell cultivation Immunostaining	Komiyama et al. (1988) [54]
	Immunoregulatory through modulation of B-cell activity		
Increased concentrations of NK cells before and decreased concentrations after periodontal therapy demonstrate that periodontitis represents an altering pathogenetic environment influencing NK cells.	Proinflammatory through upregulation and cytotoxic immunoreactive effects	Human biopsies Cell cultivation Immunostaining	Kopp et al. (1988) [55]
NK cells may have immunoregulatory functions in periodontal disease	Immunoregulatory through regulation of T-cell proliferation and suppression of B-cell immunoglobulin production	Human biopsies Cell cultivation Immunostaining	Cobb at al. (1989) [56]
Cytotoxicity was exclusively found in NK-enriched low-density large granular lymphocyte fractions after activation by LPS from **A.a.** without stimulating high levels of proliferation	Uncertain biological relevance	Lymphocytes from PB Human cell cultivation	Lindemann et al. (1989) [57]
NK cells seemed to be significantly elevated in the peripheral blood of patients with juvenile and rapidly progressive periodontitis	Proinflammatory through immuno-pathogenetic effects	Lymphocyte analysis from human PB Immunostaining	Celenligli et al. (1990) [58]
Exact nature of the role of NK cells needs further investigation	Uncertain biological relevance	Lymphocyte analysis from human PB Immunostaining	Afar at el. (1992) [59]
Leu-11b-positive cells (CD 16^+^ NK cells) appeared more frequently in severe forms of periodontitis	Proinflammatory through destructive reactions	Human biopsies Immunostaining	Fujita et al. (1992) [60]
In all of the investigated patients, cytotoxic effects of NK cells were enhanced	Proinflammatory through enhanced cytotoxic effects	Lymphocyte analysis from human PB Immunostaining	Firatli et al. (1996) [61]
Influence of NK cells and T cells in the diseased group is uncertain	Uncertain biological relevance	Lymphocytes from PB Cell cultivation Immunostaining	Mahanonda et al. (2004) [62]
**A.a.**–LPS–TLR4 interactions on dendritic cells initiate the pathway leading to the production of IFN-ɣ by NK cells in periodontitis; this results in high levels of IgG2	Proinflammatory through destructive effects via IFN-ɣ production induced by IL-12	Lymphocyte analysis from human PB Cell cultivation Immunostaining	Kikuchi et al. (2004) [63]
	Immunoregulatory through protective effects via IgG2 production		
**P.g.**–DC–NK interactions can result in reciprocal activation and increase of cytokine production by both DCs and NK cells. NK cells may provide IFN-ɣ needed to induce the **P.g.**-specific IgG2 in periodontitis	Immunoregulatory through induction of IgG2 response	Lymphocyte analysis from human PB Cell cultivation PCR	Kikuchi et al. (2005) [64]
Higher levels of CD57^+^ NK cells in tissue with periodontal disease indicate pathological progress	Proinflammatory through cytotoxic effects	Human biopsies Immunostaining	Stelin et al. (2009) [65]
Bone loss in the presence of NCR1 after infection with **F.n. F.n.** triggers the secretion of TNF-α which is dependent on NCR1 and binds directly on NCR1.	Proinflammatory through secretion of TNF-α	Induction of experimental periodontitis in mice Murine cell cultivation Immunostaining	Chaushu et al. (2012) [66]
Increased CRACC induction in aggressive periodontitis and in response to infections with **A.a.** CRACC-mediated NK cell activation could lead to an accelerated course of tissue destruction in aggressive periodontitis	Proinflammatory through CRACC induction activated by dendritic cells and subsequent IL-12 signaling	Human biopsies Transcriptomic analysis Cell cultivation	Krämer et al. (2013) [67]
Substantive increase in expression of genes related to NK cell interactions with antigen-presenting cells in periodontal tissues	Proinflammatory through cell invasion of periodontal pathogens and autoimmunity	Tissue samples of rhesus monkeys Gene expression microarray analysis	Gonzalez et al. (2014) [68]
IL-15 (bone destructive factor) activated NK cells are responsible for the induction of osteoblast apoptosis	Proinflammatory through IL-15 activation	Murine cell cultivation Immunostaining Gene expression	Takeda et al. (2014) [69]
Tissue damage is not associated with the presence or absence of different KIR genes	Uncertain biological relevance	Lymphocytes from human PB Gene analysis	Mazurek-Mochel et al. (2014) [70]
KIR presence/absence polymorphism is not a significant factor involved in the pathogenesis of periodontitis in contrast to tobacco smoking	Uncertain biological relevance	Lymphocytes from human PB Gene analysis	Mazurek-Mochol et al. (2017) [71]
Numbers of NK cells were increased in the presence of gFs. GFs support the retention and survival of NK cells by LFA-1 expression	Uncertain biological relevance	Cell cultivation Immunostaining Gene analysis	Moonen et al. (2018) [72]

#### 2.1.1. Proinflammatory Properties of NK Cells in Periodontitis

Concerning the NK cell-mediated influence on periodontitis, it is remarkable that the majority of studies [54,55,58,60,61,63,64,65,66,67,68,69] displayed proinflammatory properties of these cells through different mechanisms (Table 1). Possible models of these special characteristics were investigated in in vitro experiments [69], animal studies [66,68], and human studies through blood analysis [58,61] and tissue biopsies [54,55,60,63,64,65,67]. To provide an overview, the various proinflammatory mechanisms of NK cells interacting with soluble factors and other immune cells are illustrated in Figure 3.

By carrying out human biopsies from patients with periodontitis, cell cultivation, and immunostaining, Komiyama et al. [54], Kopp et al. [55], Fujita et al. [60], and Stelin et al. [65] demonstrated that cytotoxic reactions by NK cells cause tissue destruction. Similar observations were reported by Firatli et al. [61], Celenligil et al. [58], and Kikuchi et al. [63] when performing cell cultivation and immunostaining after collecting lymphocytes from human peripheral blood. Interestingly, Kikuchi et al. [63] demonstrated the role of NK cells in periodontitis by showing that *Aggregatibacter actinomycetemcomitans* interacts with Toll-like receptor 4 on dendritic cells and initiates pathways leading to the production of interferon gamma by NK cells. Furthermore, the authors noted that the crosstalk of dendritic cells and interferon gamma production by NK cells are dependent on interleukin 12. Krämer et al. [67] also emphasized the crosstalk of NK cells and dendritic cells through a transcriptomic analysis of human biopsies from 310 tissue samples (69 clinically healthy and 241 diseased) from 120 non-smoking patients (65 with chronic periodontitis and 55 with aggressive periodontitis). They investigated increased CD2-like receptor-activating cytotoxic cell (CRACC) induction in aggressive periodontitis in response to infections with *Aggregatibacter actinomycetemcomitans*. This CRACC-mediated NK cell activation could lead to an accelerated course of tissue destruction in aggressive periodontitis, whereas the production of interleukin 12 by dendritic cells is responsible for CRACC induction. Chaushu et al. [66], Gonzalez et al. [68], and Takeda et al. [69] utilized different animal models to examine the influence of NK cells in periodontitis. Chaushu et al. [66] created an experimental periodontal disease in mice by infection with *Fusobacterium nucleatum*. Finally, they found murine alveolar bone loss in the presence of natural cytotoxicity triggering receptor 1 (NCR1) after infection with *Fusobacterium nucleatum*. Additionally, *Fusobacterium nucleatum* bound directly to NCR1 and triggered the secretion of TNF-α as a proinflammatory mediator. Contemplating the aspect of autoimmunity, Gonzalez et al. [68] proved a substantive increase of gene expression related to NK cell interactions with antigen-presenting cells in periodontitis tissues by taking tissue samples of rhesus monkeys and conducting a microarray analysis of gene expression. They elucidated that the mechanism of NK cell-mediated pathology is an altered state of cell invasion of periodontal pathogens with a subsequent autoimmune reaction. Returning to murine cell cultivation, Takeda et al. [69] reported that interleukin 15, as a bone destructive factor, activated NK cells, leading to an induction of osteoblast apoptosis in periodontitis and rheumatoid arthritis. 

#### 2.1.2. Immunoregulatory Properties of NK Cells in Periodontitis

Conversely, other studies [54,56,63,64] have shown immunoregulatory NK cell-mediated influences on periodontitis (Table 1, Figure 4). However, these properties were exclusively found in human studies.

In addition to the proinflammatory functions, Komiyama et al. [54] and Kikuchi et al. [63] also demonstrated the protective properties of NK cells through reduced B-cell activity [54] or the production of immunoglobulin G2 (IgG2) in gingival crevicular fluid [63]. Kikuchi et al. [64] described a different role of interferon gamma as, in their opinion, interferon gamma produced by NK cells is necessary for specific IgG2 production. This process, which is induced by *Porphyromonas gingivalis*, is connected to DC–NK interactions and results in the reciprocal activation of dendritic cells and NK cells. Cobb et al. [56] examined human biopsies in their study and concluded that NK cells may have immunoregulatory functions in periodontitis through the regulation of T-cell proliferation and suppression of B-cell immunoglobulin production. 

#### 2.1.3. Uncertain Biological Relevance of NK Cells in Periodontitis

Although an evaluation of a definite role for NK cells in periodontitis might be possible, there exist publications investigating various aspects of NK cells that remain uncertain in their outcome [57,59,62,70,71,72] (Table 1, Figure 4). Five of these mentioned studies [57,59,62,70,71] were performed by a human biopsy and analysis of peripheral blood lymphocytes, and only one was conducted by in vitro cell cultivation [72]. For instance, Afar et al. [59] stated that NK cells might be relevant for chronic periodontitis, whereas the exact nature of the role of NK cells is uncertain. Similar observations were found by Mazurek-Mochol et al. [70,71], who admitted that the presence or absence of killer cell immunoglobulin-like receptor (KIR) genes was not associated with parameters such as an increased pocket depth and clinical attachment loss corresponding to the severity of the disease.

### 2.2. NKT Cell-Mediated Mechanisms in Periodontitis 

NKT cell-mediated mechanisms were evaluated in four different investigations, which are represented in the following overview (Table 2). 

**Table 2 ijms-21-09766-t002:** NKT cell-mediated mechanisms in periodontitis. NKT—natural killer T cell; A.a.—*Aggregatibacter actinomycetemcomitans*; P.g—*Porphyromonas gingivalis*; CD—cluster of differentiation; and IFN-ɣ—interferon gamma.

Mechanism	Evaluation of the Influence	Study Design	Reference
Invariant NKT cells infiltrating periodontal lesions seem to have a downregulating role of autoimmune responses	Immunoregulatory through association with CD1d(+) cells	Human biopsies Lymphocytes from human peripheral blood Gene analysis Immunostaining	Yamazaki et al. (2001) [73]
NKT cells can provide direct help for B-cell proliferation and antibody production against autoimmune reactions	Immunoregulatory through activation by CD1d-expressing B cells	Human biopsies Immunostaining	Amanuma et al. (2006) [74]
Type I NKT cell activation and IFN-ɣ secretion possibly aggravate tissue destruction in aggressive periodontitis	Proinflammatory through activation of type I NKT cells with subsequent production of IFN-ɣ triggered by A.a.	Human biopsies Transcriptomic analysis Murine cell cultivation	Nowak et al. (2013) [75]
Activation of NKT cells promoted a systematic inflammatory response. **P.g.** induced alveolar bone resorption via NKT cell activation	Proinflammatory through T-cell and B-cell activation mediated by NKT cells via IFN-ɣ	Murine periodontal infection Murine biopsy, blood, and gene analysis	Aoki-Nonaka et al. (2014) [76]

#### 2.2.1. Immunoregulatory Properties of NKT Cells in Periodontitis

Performing human biopsies and immunostaining [74] in combination with an analysis of lymphocytes from peripheral blood and gene analysis [73], Yamazaki et al. and Amanuma et al. concluded that NKT cells infiltrating periodontal lesions seem to have a downregulating role in autoimmune responses via activation by CD1d-expressing B cells. This could lead to the production of antibodies in autoimmune reactions in periodontitis [74]. 

#### 2.2.2. Proinflammatory Properties of NKT Cells in Periodontitis

In contrast, Nowak et al. [75] investigated different influences of invariant NKT cells on chronic and aggressive periodontitis and stated that type I NKT cell activation and interferon gamma secretion possibly aggravate tissue destruction in aggressive periodontitis. According to their human study, *Aggregatibacter actinomycetemcomitans*, unlike *Porphyromonas gingivalis*, triggered the production of interferon gamma via binding to Toll-like receptors of dendritic cells. Aoki-Nonaka et al. [76] depicted proinflammatory aspects in local tissue and above all, systematic influences. In their experimental model of murine periodontal infection, T-cell and B-cell activation was mediated by NKT cells via interferon gamma [76]. They highlighted that the activation of NKT cells promoted a systemic inflammatory response and *Porphyromonas gingivalis* interacting with local immunity induced alveolar bone resorption [76]. The authors assumed a T helper cell shift from type I toward type II by providing evidence of interleukin 4 and 10 production [76].

NKT cell-mediated mechanisms in periodontitis are illustrated in Figure 5. According to the reports, their role is contradictious, as the respective studies have reported on immunoregulatory [73,74] and proinflammatory properties [75,76]. 

### 2.3. Combined NK Cell-/NKT Cell-Mediated Mechanisms in Periodontitis

Interestingly, two human studies (Table 3) reporting combined NK cell-/NKT cell-mediated mechanisms in periodontitis [77,78] utilized the attraction of mononuclear lymphocytes by a chemokine gradient, which is often seen in periodontitis created by gingival fibroblasts. Hosokawa et al. [77] and Muthukuru et al. [78] performed human biopsies and subsequent cell cultivation, concluding that both cell subsets have proinflammatory influences in periodontal disease through the production of interferon gamma [77] or downregulation of T helper cells [78].

**Table 3 ijms-21-09766-t003:** Combined NK cell-/NKT cell-mediated mechanisms in periodontitis. NK—natural killer cell; NKT—natural killer T cell; gF—gingival fibroblast; CXCL—CXC-motif ligand chemokine; and IFN-ɣ—interferon gamma.

Mechanism	Evaluation of the Influence	Study Design	Reference
CXCL 16 produced by human gFs in diseased periodontal tissues controls the migration of NK and NKT cells, leading to bone resorption	Proinflammatory through IFN-ɣ and further attraction by chemokines	Human biopsies Cell cultivation Gene analysis	Hosokawa et al. (2007) [77]
Increased number of NK and NKT cells in tissues from patients with chronic periodontitis	Proinflammatory through downregulation of T-helper cells	Human biopsies Cell cultivation Immunostaining	Muthukuru et al. (2012) [78]

## 3. Discussion

The aim of this article was to evaluate the influences of natural killer cells and natural killer T cells on periodontal disease with a systematic review of the literature. The focus was directed towards publications investigating the immunologically cell-mediated mechanisms contributing to pathogenesis or reestablished oral health. 

### 3.1. Role of NK Cells in Periodontitis

In oral health, NK cells, as members of the innate immunity, fulfill important duties when it comes to a possible threat for the host. One crucial factor is the differentiation between self and foreign antigens by the presence or absence of the major histocompatibility complex class I (MHC-I) molecules on the surface of the presenting cell. Based on the fact that NK cells recognize MHC-I molecules on the surface of healthy cells, they will be inhibited by the receptors, rather than activated. However, if normal/healthy cells are transformed, e.g., by bacteria, viral infections, or tumors, NK cells will be activated by the reduction of MHC-I molecules on the surfaces of these transformed cells [14]. Stimulation by activatory receptors leads to enhanced killing of the target or cytokine production, or to both characteristics [79,80,81,82]. Considering the influences of NK cell-mediated mechanisms in dental inflammatory disease such as periodontitis found in the literature in this research, several proinflammatory NK cell-mediated mechanisms could be identified in in vitro experiments [69], animal studies [66,68], and human studies [54,55,58,60,61,63,64,65,67]. Among these inflammation-promoting effects, the production of interferon gamma and crosstalk with dendritic cells via interleukin 12 [63], the activation through interleukin 15 as a bone destructive factor [69], the production of tumor necrosis factor alpha [66], autoimmune effects [68], cytotoxic effects [55,60,61,65], immunopathogenic effects [54,58], and crosstalk with dendritic cells in combination with the activation of cytotoxic properties [67] could be identified. These findings are consistent with the reviews of other authors citing proinflammatory influences of NK cells in periodontitis [14,18,83,84]. Remarkably, the reviews of Gonzales et al. [84] and Meyle et al. [83] mixed up NK cells with NKT cells, whereas both subsets presented separate cell entities with certain differences. Interestingly, the production of interferon gamma is one central aspect concerning the NK cell-mediated mechanism in periodontitis, as increased levels of interferon gamma are predictive for the severity of periodontal disease [85]. Similar findings were observed in other pathologies such as Type I diabetes [25,26] and atherosclerosis [79,80], where interferon gamma, which is produced by NK cells, triggered an excessive, uncontrolled, and unresolved immune response in those diseases. Crosstalk with dendritic cells and the subsequent activation of NK cells via the production of interleukin-12 was also one crucial element accelerating tissue breakdown in periodontitis, which was described for artherosclerotic plaque formation by Bonaccorsi et al. [37].

Moreover, the aim of this article was to investigate possible models of the physiological functions of NK cells in order to reestablish oral health. Regarding the eligible reports, different immunoregulatory mechanisms, such as the regulation of T-cell proliferation, suppression of B cells producing nonspecific immunoglobulins [56], stimulation of the production of specific IgG2 antibodies [63,64], and modulation of B-cell activity [54], were exclusively identified in human studies. This is notable as a nonspecific antibody response induced by B cells/plasma cells is not protective [86], plasma cell-secreted antibodies might worsen inflammatory disease, and B and T cells participate in bone resorption via secretion of the receptor activator of the NF-κB ligand (RANKL) [87]. Similarly, Parisi et al. [18] and Wilensky et al. [14] cited the possible immunoregulatory role of NK cells in periodontitis in their reviews; however, they considered the role of NK cells as primarily regulatory, requiring fewer cells. Additionally, an altered state of NK cells was discovered for different types of cancer as the tumor microenvironment led to reduced tumor cell lysis through a decreased expression of activatory receptors and the release of perforin and granzyme [18].

Regrettably, six eligible reports, including five human studies [57,59,62,70,71] and one in vitro study [72], failed to show any relationship between NK cells and periodontitis. Whilst the authors supposed that NK cells might be involved in the etiology of periodontal disease, the exact nature of the role of NK cells is uncertain [57,59,62,70,71,72]. 

Considering oral diseases, it is worth mentioning that there exists one available study dealing with the influence of NK cells on peri-implantitis [88], which rather similar oral inflammation when compared to periodontitis [89], and none concerning NKT cells. De Araujo et al. [88] analyzed the expression of CD15 (a neutrophil marker), CD57 (a natural killer cell marker), and hypoxia inducible factor (HIF)-1α (a hypoxia marker) in biopsies of patients with peri-implantitis compared to healthy patients. Higher levels of CD15 and HIF-1α were seen in the peri-implantitis group; however, no significant differences of CD57 levels were observed between groups. The authors attributed this observation to the cytotoxic activity of NK cells towards virus-infected cells, whereas peri-implantitis is a bacteria-mediated disease [88]. Taking the mentioned results and the preliminary reports into account, it can be assumed that NK cells seem to have proinflammatory effects on periodontitis, rather than immunoregulatory or no effects (Figure 2 and Figure 3). The limitations within the provided findings are that not all reports included human studies with a further analysis of biopsies of the diseased area. Indeed, it is important to note that parts of the cited results were only discovered in animals and in vitro experiments, leaving some doubts on the transferability of NK cell-mediated mechanisms to human periodontitis. Therefore, studies with human biopsies from diseased patients in comparison with healthy controls and subsequent transcriptomic analysis should be preferred for the prediction of NK cell-mediated mechanisms in human periodontitis. Moreover, future studies should also focus on NK cell-mediated mechanisms in peri-implantitis to figure out further similarities or differences to periodontal disease.

### 3.2. Role of NKT Cells in Periodontitis

NKT cells combine innate and adaptive properties, as they express an antigen-specific T-cell receptor (TCR), but only have a strongly limited repertoire of antigen reactions mediated by these receptors. Furthermore, these receptors display similarities to pattern-recognition receptors of innate immune cells. NKT cells could not develop an immunological memory and showed characteristics such as the rapid elicitation of effector functions, which might be deleterious, or a protective tendency for autoimmunity [39]. NKT cells are directly activated by the binding of lipid or glycolipid antigens with the CD1d receptor on the antigen-presenting cell (APC) and the semi-invariant TCR on NKT cells or indirectly by microorganisms that do not contain cognate glycolipid antigens [90,91].

The findings of this review for NKT cell-mediated mechanisms in periodontitis were contradictory. Two authors reported [75,76] proinflammatory effects similar to NK cell-mediated pathologies in human/animal studies on the production of interferon gamma as a primary source for the activation of other cells. On the other hand, according to the observations of human biopsies and blood analysis of Yamazaki et al. [73] and Amanuma et al. [74], NKT cells possess immunoregulatory properties by interacting with CD1d-expressing B cells [73], and subsequent downregulation of the autoimmune response through antibody production [74]. These data were also depicted in the reviews of Gonzales et al. [84] and Meyle [83], although they did not differentiate between NK cells and NKT cells. Similarities could be found when comparing the influence of NKT cells on other human pathologies with periodontitis, as they can exhibit proinflammatory properties by a burst of cytokines and anti-inflammatory properties by immunoregulation proven by diverse studies [45,46,47,48,49,50,51,52].

Moreover, studies should differentiate between the sublineages of NKT cells as there exist iNKT cells (type I) and vNKT cells (type II), which play a cross-regulating role with iNKT cells [42]. Only the report of Nowak et al. [75] gave concrete information concerning this issue, as they depicted iNKT cells. 

Regarding the reported facts concerning the NKT cell-mediated mechanisms in periodontitis, it can be assumed that NKT cells could have proinflammatory and immunoregulatory influences in periodontitis that are similar to other human pathologies (Figure 4). Comparing the reports, at least three out of four contain human studies with a further examination of biopsies and subsequent transcriptomic analysis, delivering more transferable results than in vitro or animal models. Nevertheless, as only four reports were eligible, the data are limited and further research focusing on NKT cells and periodontitis is needed. Additionally, studies considering peri-implantitis should be performed. 

### 3.3. Combined Role of NK/NKT Cells in Periodontitis

In this review, two investigations of human biopsies dealing with combined NK cell-/NKT cell- mediated mechanisms in periodontitis could be found [77,78]. Both utilized the attraction of mononuclear lymphocytes by a chemokine gradient created by gingival fibroblasts, which is a prerequisite for the formation of inflammation in periodontitis. Interestingly, the combination of the two cell subsets seems to have an accumulative destructive effect through the production of interferon gamma and downregulation of the T helper cell response. The latter issue is of particular importance as the traditional role model of T helper cells, namely T helper cells type I and II, in periodontitis, provides discrepancies [92]. 

### 3.4. Role of Different Bacteria 

Contemplating the study designs of the eligible studies, it should be noted that human biopsies and animal/in vitro experiments with cell cultures, immunostaining, or transcriptomic analysis in interaction with periodontal pathogens of the biofilm, such as *Porphyromonas gingivalis*, *Aggregatibacter actinomycetemcomitans*, and *Fusobacterium nucleatum*, were performed. *Porphyromonas gingivalis* can be regarded as a ‘keystone pathogen’ able to convert the homeostasis towards dysbiosis in susceptible hosts, whereas *Aggregatibacter actinomycetemcomitans* and *Fusobacterium nucleatum* are orchestrated as so-called ‘pathobionts’ by the ‘keystone species’, causing tissue destruction [7].

In sum, it is important to note that some experimental models of the cited results were only conducted in vitro and not in vivo. Future studies should concentrate on human biopsies taken from the gingivae of periodontitis patients with transcriptomic tissue analysis of, e.g., NKT cell components such as T-cell receptors. 

## 4. Materials and Methods 

### 4.1. Study Design 

For preparing this systematic review, the PRISMA Statement was utilized as a reporting item [53].

### 4.2. Search Strategy

A literature search of MEDLINE (PubMed) was carried out using the following predetermined strategy: ((“killer cells, natural”[MeSH Terms] OR (“killer”[All Fields] AND “cells”[All Fields] AND “natural”[All Fields]) OR “natural killer cells”[All Fields] OR (“natural”[All Fields] AND “killer”[All Fields] AND “cells”[All Fields])) OR (“killer cells, natural”[MeSH Terms] OR (“killer”[All Fields] AND “cells”[All Fields] AND “natural”[All Fields]) OR “natural killer cells”[All Fields] OR (“nk”[All Fields] AND “cells”[All Fields]) OR “nk cells”[All Fields]) OR (“natural killer t-cells”[MeSH Terms] OR (“natural”[All Fields] AND “killer”[All Fields] AND “t-cells”[All Fields]) OR “natural killer t-cells”[All Fields] OR “natural killer t cells”[All Fields]) OR (“nk”[All Fields] AND (“t-lymphocytes”[MeSH Terms] OR “t-lymphocytes”[All Fields] OR “t cells”[All Fields]))) AND ((“periodontics”[MeSH Terms] OR “periodontics”[All Fields] OR “periodontology”[All Fields]) OR (“periodontitis”[MeSH Terms] OR “periodontitis”[All Fields]). The search of the latest literature from the time period 1988–2020 resulted in 89 publications.

### 4.3. Study Selection

To prove the eligibility of the studies, the following inclusion and exclusion criteria were formulated:Original publications regarding animals, humans (peripheral lymphocyte analysis and human biopsies), and in vitro studies (cell cultivation) were relevant;The results had to include the term ‘marginal periodontitis’;Reviews were excluded and were only used for discussing the outcome;Publications containing titles of other diseases, such as peri-apical periodontitis, saliva-associated diseases, hypophosphatasia, hypercholesterolemia, cancer, psychological stress disorders, and cranio-facial syndromes, were excluded;Abstracts fulfilling the inclusion criteria and dealing with different immune cells, but not with NK cells and NKT cells, were extracted.

### 4.4. Data Collection Process

Eligible studies (human/animal/in vitro) were screened and data extraction sheets containing the publication, abstract, study subject, cell subset (NK cells/NKT cells/both), study design, study duration, outcome, and conclusion were created.

## 5. Conclusions

Considering the influence of natural killer cells and natural killer T cells on periodontal disease, it can be concluded that the majority of citied studies investigated NK cells, rather than NKT cells. As demonstrated, NK cells and NKT cells contribute to proinflammatory reactions via cytotoxic reactions, the production of chemokines and cytokines, the modulation of B and T cells, the upregulation of autoimmunity, and crosstalk with dendritic cells. Conversely, NK cells and NKT cells can also have immunoregulatory impacts on B cells, T cells, and dendritic cells via the downregulation of autoimmunity, in order to reestablish oral health. Remarkably, there are more proinflammatory models regarding the combination of NK cell- and NKT cell-mediated mechanisms in periodontitis leading to dysbiosis than immunoregulatory ones contributing to eubiosis. These findings should lead the way to further research on the precise immune responses in periodontitis, in order to deduce a better understanding of oral diseases and to create, in addition to the conventional anti-bacterial-targeted therapy of periodontitis, novel anti-inflammatory therapies. Regarding anti-inflammatory approaches, an innovative therapy model was presented by Hasturk et al. [93] with pro-resolution lipid mediators following microbial challenges. In their observation, the local and systemic inflammatory response could be regulated by the small-lipid molecules of lipoxin, having a direct influence on the formation of dental biofilm. Due to the limited data, future studies should focus on NKT cells. It is relevant to identify the exact mechanism of NK/NKT cell-mediated immunological effects in interaction/crosstalk with other immune cells.

## Figures and Tables

**Figure 1 ijms-21-09766-f001:**
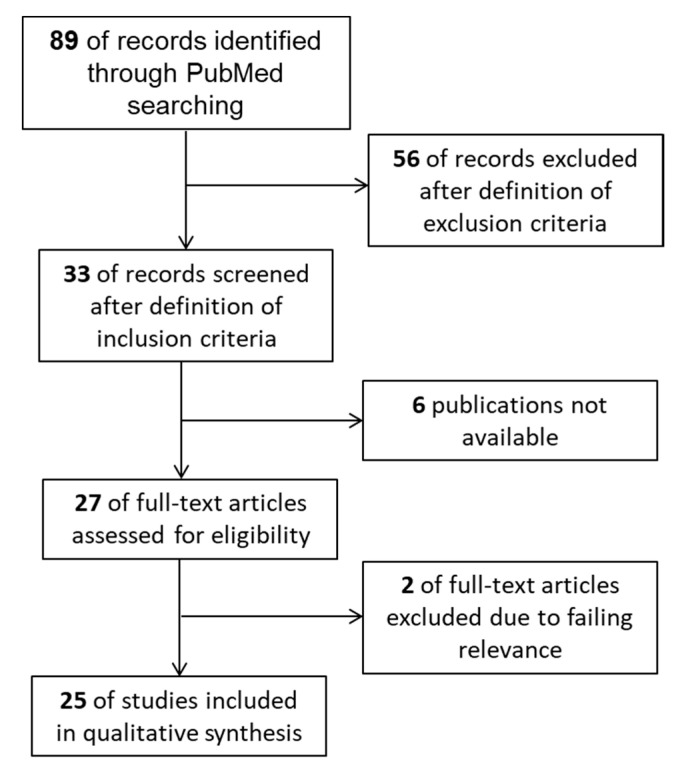
Flow diagram containing the process of study selection.

**Figure 2 ijms-21-09766-f002:**
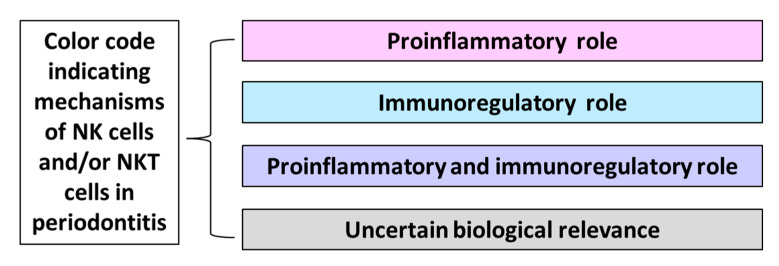
Color code indicating mechanisms of natural killer (NK) cells and/or natural killer T (NKT) cells in periodontitis.

**Figure 3 ijms-21-09766-f003:**
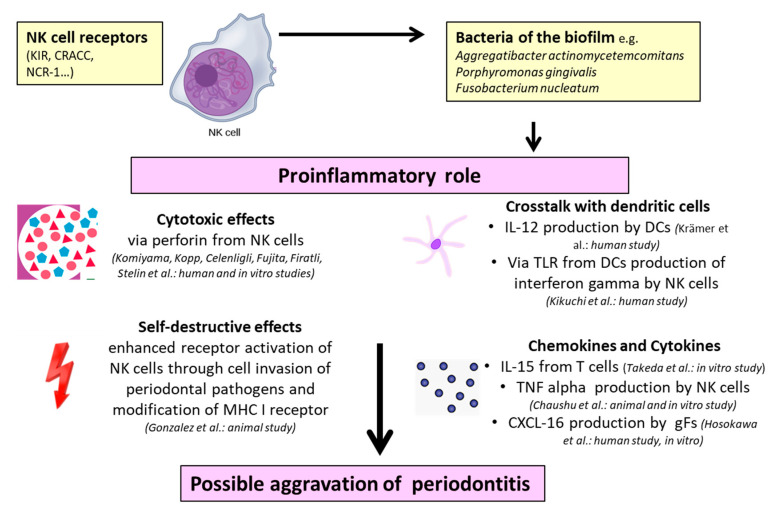
Proinflammatory properties of NK cells in periodontitis: The references and experimental study model are given in parentheses (human, animal, and in vitro). NK cell—natural killer cell; DC—dendritic cell; gF—gingival fibroblast; KIR—killer cell immunoglobulin-like receptor; CRACC—CD2-like receptor-activating cytotoxic cell; NCR-1—natural cytotoxicity triggering receptor 1; IL—interleukin; TLR—Toll-like receptor; MHC—major histocompatibility complex; TNF-α—tumor necrosis factor alpha; and CXCL—CXC-motif ligand chemokine.

**Figure 4 ijms-21-09766-f004:**
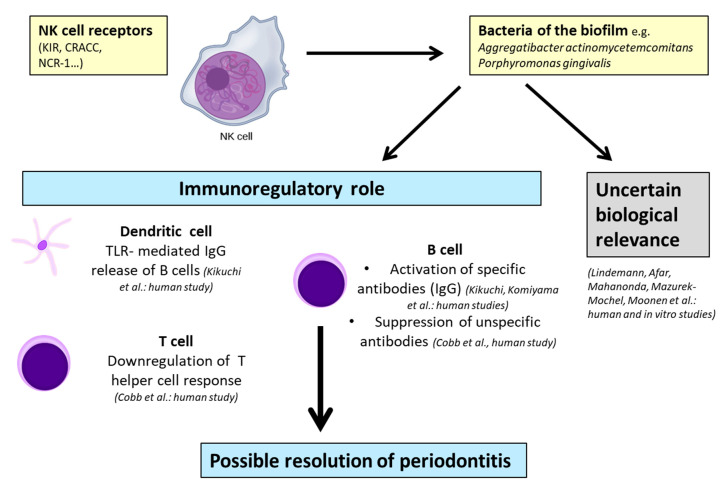
Possible models of immunoregulatory properties and uncertain biological relevance of NK cells in periodontitis: The references and experimental study model are given in parentheses (human, animal, and in vitro). NK cell—natural killer cell; KIR—killer cell immunoglobulin-like receptor; CRACC, CD2-like receptor-activating cytotoxic cell; NCR-1—natural cytotoxicity triggering receptor 1; IgG—immunoglobulin G; and TLR—Toll-like receptor.

**Figure 5 ijms-21-09766-f005:**
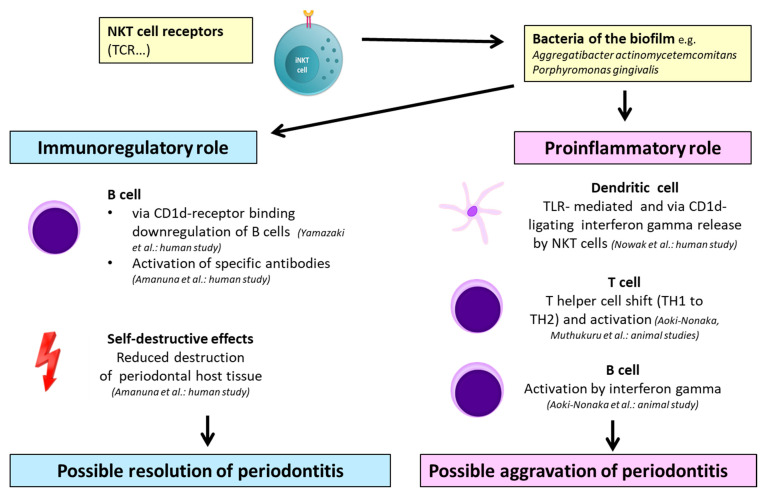
Possible models of NKT cell-mediated mechanisms in periodontitis: The references and experimental study model is given in parentheses (human, animal, and in vitro). NK—natural killer cell; NKT—natural killer T cell; DC—dendritic cell; iNKT cell—invariant natural killer T cell; TH—T helper cell; TCR—T-cell receptor; KIR—killer cell immunoglobulin-like receptor; CRACC—CD2-like receptor-activating cytotoxic cell; NCR-1—natural cytotoxicity triggering receptor 1; IL—interleukin; TLR—Toll-like receptor; and CD—cluster of differentiation.

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
