# Peer review of "Influence of Natural Killer Cells and Natural Killer T Cells on Periodontal Disease: A Systematic Review of the Current Literature"

_ijms, 2020, doi:10.3390/ijms21249766_

Round 1

Reviewer 1 Report

Reviewer’s comments

Seidel, at el wrote a meta-analytic review of 26 studies on the roles of mainly NK cells in periodontitis. It is interesting to know the roles of NK cells in oral/dental diseases. The review is well organised and written.

Major point

  1. Among 26 studies which the authors meta-analysed, 25 focus on NK cells in periodontitis. NKT cells were studied in only 1 paper, and so peri-implantitis was. Wouldn’t it be fairer if the review focuses only NK cells in periodontitis? I think NKT cells and peri-implantitis could be briefly mentioned in Discussion.

Minor points

  1. Table 1 is color-coded, but the color-codes are not explained.
  2. There are some minor typos and mis-spells, for example, in Page 13 Line 351, ‘typ’ should be ‘type’.
  3.  

Author Response

Dear Reviewer 1,

Thank you very much for you positive feedback and helpful comments. Please find enclosed our manuscript ijms-1036353-rev including the requested major and minor changes. All changes in the manuscript are presented in word correction mode highlighted in red letters. Below is a point-by-point response to your comments.

Major point

  1. Among 26 studies which the authors meta-analysed, 25 focus on NK cells in periodontitis. NKT cells were studied in only 1 paper, and so peri-implantitis was. Wouldn’t it be fairer if the review focuses only NK cells in periodontitis? I think NKT cells and peri-implantitis could be briefly mentioned in Discussion.

We really appreciate your comment. Therefore, we decided to exclude the item peri-implantitis from our research, as there was only one study dealing with this issue. Moreover, we agree with the reviewer to briefly mention the role of NK cells in peri-implantitis in the discussion of this manuscript. Concerning the influence of NKT cells on periodontitis, we finally found six eligible studies, not one study. For this reason we would suggest not only to focus on NK and periodontitis, but also to investigate NKT cell-mediated mechanisms in periodontitis.

Minor points

  1. Table 1 is color-coded, but the color-codes are not explained.

Thank you for this constructive annotation. We added a figure (Figure 2, page 5 line 150-153) explaining the color code seen in the tables and figures throughout the manuscript.

  1. There are some minor typos and mis-spells, for example, in Page 13 Line 351, ‘typ’ should be ‘type’.

We apologize for the mis-spells and typos. To our best knowledge, we corrected all false typos and spellings including “typ” into “type”.

We are confident that our changes improved the manuscript and help to enhance comprehensibility and readability. We hope that the manuscript in its revised form is now acceptable for publication. Please don’t hesitate to contact us for further questions.

With best regards we remain

Andreas Seidel, Corinna Seidel and Helga Schmetzer

Reviewer 2 Report

The review summarized 26 studies related to innate immunity and periodontitis, specifically NK cell and NKT cell. The study gives very limited insights and further directions in this field. The description and figure demonstration is highly redundant through the manuscript.  

  1. The colors of tables and figures are not friendly for reading.
  2. Line 55: The introductions of NK cells and NKT cells are not sufficient.
  3. Figure 2: ‘cytotoxic reactions’ is not the standard term in the field. What ‘autoimmunity’ exactly mean and how it integrates in the figure? And the descriptions of studies are not accurate, for instance, how evasion of pathogen and MHC molecules cause autoimmunity? What exactly mean “upregulation of autoimmunity” in the figure legend? Also, indicating references in all figures may help readers.
  4. Line 197: what do authors mean ‘unclear role’? or do they mean ‘biological relevance needs further investigations’? Please rephrase the term.
  5. The descriptions in main text, figure legends, and discussion are highly redundant. 
  6. The Figure 5 is a simply copy of Figure 2/3/4. Please condense the figures.

Author Response

Dear Reviewer 2,

Thank you very much for you positive feedback and helpful comments. Please find enclosed our manuscript ijms-1036353-rev including the requested major and minor changes. All changes in the manuscript are presented in word correction mode highlighted in red letters. Below is a point-by-point response to your comments.

  1. The colors of tables and figures are not friendly for reading.

We have to thank for this annotation. To improve readability of tables and figures, we changed the colors into more eye-friendly softer shades (Figure 2-5, Table 1-3).

  1. Line 55: The introductions of NK cells and NKT cells are not sufficient.

Thank you for your remarks. Concerning this point, we provided further information on the NK- and NKT cell- mediated pathologies. Therefore, we explained the role of NK cells in different disease like cancer, diabetes mellitus and atherosclerosis (page 2, line 71-90). Additionally, we depicted the influence of NKT cells on autoimmune diseases, leukemia, atherosclerosis, arthritis and biliary cirrhosis. In this context, the following citations were added (page 3, line 102-111):

Clayton A, Mitchell JP, Court J, Linnane S, Mason MD, Tabi Z. Human tumor-derived exosomes down-modulate NKG2D expression. J Immunol Baltim Md 1950. 2008 Jun 1;180(11):7249–58.

Clayton A, Tabi Z. Exosomes and the MICA-NKG2D system in cancer. Blood Cells Mol Dis. 2005 Jun;34(3):206–13.

Muller L, Mitsuhashi M, Simms P, Gooding WE, Whiteside TL. Tumor-derived exosomes regulate expression of immune function-related genes in human T cell subsets. Sci Rep. 2016 Feb 4;6:20254.

Greening DW, Gopal SK, Xu R, Simpson RJ, Chen W. Exosomes and their roles in immune regulation and cancer. Semin Cell Dev Biol. 2015 Apr;40:72–81.

Webber J, Yeung V, Clayton A. Extracellular vesicles as modulators of the cancer microenvironment. Semin Cell Dev Biol. 2015 Apr;40:27–34.

Baginska J, Viry E, Paggetti J, Medves S, Berchem G, Moussay E, et al. The critical role of the tumor microenvironment in shaping natural killer cell-mediated anti-tumor immunity. Front Immunol. 2013 Dec 25;4:490.

Eisenbarth GS. Type I diabetes mellitus. A chronic autoimmune disease. N Engl J Med. 1986 May 22;314(21):1360–8.

Kelly MA, Rayner ML, Mijovic CH, Barnett AH. Molecular aspects of type 1 diabetes. Mol Pathol MP. 2003 Feb;56(1):1–10.

Baxter AG, Smyth MJ. The role of NK cells in autoimmune disease. Autoimmunity. 2002 Feb;35(1):1–14.

Rodacki M, Milech A, de Oliveira JEP. NK cells and type 1 diabetes. Clin Dev Immunol. 2006 Dec;13(2–4):101–7.

Rodacki M, Svoren B, Butty V, Besse W, Laffel L, Benoist C, et al. Altered natural killer cells in type 1 diabetic patients. Diabetes. 2007 Jan;56(1):177–85.

Bobryshev YV, Lord RSA. Identification of natural killer cells in human atherosclerotic plaque. Atherosclerosis. 2005 Jun;180(2):423–7.

Whitman SC, Rateri DL, Szilvassy SJ, Yokoyama W, Daugherty A. Depletion of natural killer cell function decreases atherosclerosis in low-density lipoprotein receptor null mice. Arterioscler Thromb Vasc Biol. 2004 Jun;24(6):1049–54.

Allavena P, Bianchi G, Zhou D, van Damme J, Jílek P, Sozzani S, et al. Induction of natural killer cell migration by monocyte chemotactic protein-1, -2 and -3. Eur J Immunol. 1994 Dec;24(12):3233–6.

Umehara H, Bloom ET, Okazaki T, Nagano Y, Yoshie O, Imai T. Fractalkine in vascular biology: from basic research to clinical disease. Arterioscler Thromb Vasc Biol. 2004 Jan;24(1):34–40.

Chistiakov DA, Sobenin IA, Orekhov AN, Bobryshev YV. Dendritic cells in atherosclerotic inflammation: the complexity of functions and the peculiarities of pathophysiological effects. Front Physiol. 2014;5:196.

Mallat Z, Corbaz A, Scoazec A, Graber P, Alouani S, Esposito B, et al. Interleukin-18/interleukin-18 binding protein signaling modulates atherosclerotic lesion development and stability. Circ Res. 2001 Sep 28;89(7):E41-45.

Uyemura K, Demer LL, Castle SC, Jullien D, Berliner JA, Gately MK, et al. Cross-regulatory roles of interleukin (IL)-12 and IL-10 in atherosclerosis. J Clin Invest. 1996 May 1;97(9):2130–8.

Bonaccorsi I, De Pasquale C, Campana S, Barberi C, Cavaliere R, Benedetto F, et al. Natural killer cells in the innate immunity network of atherosclerosis. Immunol Lett. 2015 Nov;168(1):51–7.

Van Kaer L. NKT cells: T lymphocytes with innate effector functions. Curr Opin Immunol. 2007 Jun;19(3):354–64.

Nowak M, Stein-Streilein J. Invariant NKT cells and tolerance. Int Rev Immunol. 2007 Apr;26(1–2):95–119.

Pillai AB, George TI, Dutt S, Teo P, Strober S. Host NKT cells can prevent graft-versus-host disease and permit graft antitumor activity after bone marrow transplantation. J Immunol Baltim Md 1950. 2007 May 15;178(10):6242–51.

Boeck CL, Amberger DC, Doraneh-Gard F, Sutanto W, Guenther T, Schmohl J, et al. Significance of Frequencies, Compositions, and/or Antileukemic Activity of (DC-stimulated) Invariant NKT, NK and CIK Cells on the Outcome of Patients With AML, ALL and CLL. J Immunother Hagerstown Md 1997. 2017 Aug;40(6):224–48.

Terabe M, Berzofsky JA. The role of NKT cells in tumor immunity. Adv Cancer Res. 2008;101:277–348.

Coppieters K, Van Beneden K, Jacques P, Dewint P, Vervloet A, Vander Cruyssen B, et al. A single early activation of invariant NK T cells confers long-term protection against collagen-induced arthritis in a ligand-specific manner. J Immunol Baltim Md 1950. 2007 Aug 15;179(4):2300–9.

Joyce S, Van Kaer L. Invariant natural killer T cells trigger adaptive lymphocytes to churn up bile. Cell Host Microbe. 2008 May 15;3(5):275–7.

  1. Figure 2: ‘cytotoxic reactions’ is not the standard term in the field. What ‘autoimmunity’ exactly mean and how it integrates in the figure? And the descriptions of studies are not accurate, for instance, how evasion of pathogen and MHC molecules cause autoimmunity? What exactly mean “upregulation of autoimmunity” in the figure legend? Also, indicating references in all figures may help readers.

Thank you for valuable annotations. In former Figure 2 (now Figure 3), we replaced the term ‘cytotoxic reactions’ by the expression ‘cytotoxic effects’ as it characterizes the activation and immunological behavior of NK cells more precisely. Moreover, we think that ‘autoimmunity’ in Figure 3 means a self-destructive effect leading to damage of host tissue mediated by NK cells. Therefore, we replaced the term ‘autoimmunity’ with the expression ‘self-destructive effects’. Here, enhanced NK cell activation was induced by cell invasion of periodontal pathogens and modification of MHC-I receptor leading to increased killing of gingival host tissue. The modification of MHC-1 receptors of gingival tissue cells accelerates the proliferation and the killing of this tissue by NK cells, as NK cells could no longer distinguish between host and modified cells. Based on upregulation of these self-destructive effects mediated by NK cells this results in tissue breakdown and aggravation of periodontitis.

To help the reader, we added the references in all Figures (Figure 3,4,5).

  1. Line 197: what do authors mean ‘unclear role’? or do they mean ‘biological relevance needs further investigations’? Please rephrase the term.

Special thanks concerning this point. We rephrased the term ‘unclear role’ of NK cells by the expression ‘uncertain biological relevance’ in the whole manuscript, as we also believe that this term fits more to the reported investigations (table 1, figure 3).

  1. The descriptions in main text, figure legends, and discussion are highly redundant.

We are grateful for this remark. Therefore, we shortened the descriptions in the main text, figure legends and discussion.

  1. The Figure 5 is a simply copy of Figure 2/3/4. Please condense the figures.

We appreciate your point of view and removed former figure 5. Furthermore, we condensed the figures by shortening figure legends (now Figure 3,4,5).

We are confident that our changes improved the manuscript and help to enhance comprehensibility and readability. We hope that the manuscript in its revised form is now acceptable for publication. Please don’t hesitate to contact us for further questions.

With best regards we remain

Andreas Seidel, Corinna Seidel and Helga Schmetzer

Round 2

Reviewer 2 Report

'Unclear role' still appears in current Fig.4. Please change to 'uncertain biological relevance'.

Author Response

Dear Reviewer 2,

Thank you very much for you positive feedback and helpful comments. Please find enclosed our manuscript ijms-1036353-(1) including the requested major and minor changes. All changes in the manuscript are presented in word correction mode highlighted in red letters. Below is a point-by-point response to your comments.

  1. 'Unclear role' still appears in current Fig.4. Please change to 'uncertain biological relevance'.

We have to thank you for this annotation and removed “unclear role” from Figure 4. Moreover, we removed the work “unclear” from the explanation of the color code (line 141) and from Table 1 (Afar et al. 1992).

Moreover, we corrected some small typing and formatting errors.

We are confident that our changes improved the manuscript! We hope that the manuscript in its revised form is now acceptable for publication. Please don’t hesitate to contact us for further questions.

With best regards we remain

Andreas Seidel, Corinna Seidel and Helga Schmetzer
